# PD-1 blockade potentiates HIV latency reversal ex vivo in CD4$^+$ T cells from ART-suppressed individuals

Rémi Fromentin[1], Sandrina DaFonseca[2], Cecilia T. Costiniuk[3], Mohamed El-Far[1], Francesco Andrea Procopio[4], Frederick M. Hecht [5], Rebecca Hoh[5], Steven G. Deeks [5], Daria J. Hazuda[6], Sharon R. Lewin[7,8], Jean-Pierre Routy [3], Rafick-Pierre Sékaly[9] & Nicolas Chomont [1,10]

HIV persists in latently infected CD4$^+$ T cells during antiretroviral therapy (ART). Immune checkpoint molecules, including PD-1, are preferentially expressed at the surface of persistently infected cells. However, whether PD-1 plays a functional role in HIV latency and reservoir persistence remains unknown. Using CD4$^+$ T cells from HIV-infected individuals, we show that the engagement of PD-1 inhibits viral production at the transcriptional level and abrogates T-cell receptor (TCR)-induced HIV reactivation in latently infected cells. Conversely, PD-1 blockade with the monoclonal antibody pembrolizumab enhances HIV production in combination with the latency reversing agent bryostatin without increasing T cell activation. Our results suggest that the administration of immune checkpoint blockers to HIV-infected individuals on ART may facilitate latency disruption.

[1] Centre de Recherche du CHUM, Montréal H2X0A9 QC, Canada. [2] Caprion Biosciences Inc., Montréal H2X3Y7 QC, Canada. [3] Chronic Viral Illness Service and Division of Hematology, Research Institute, McGill University Health Centre, Montréal H4A3J1 QC, Canada. [4] Service of Immunology and Allergy, Lausanne University Hospital, University of Lausanne, 1011 Lausanne, Switzerland. [5] Department of Medicine, University of California San Francisco, San Francisco, CA 94115, USA. [6] Infectious Disease, Merck Research Laboratories, West Point, PA 19486, USA. [7] The Peter Doherty Institute for Infection and Immunity, The University of Melbourne and Royal Melbourne Hospital, Melbourne, VIC 3000, Australia. [8] Department of Infectious Diseases, Alfred Health and Monash University, Melbourne, Australia. [9] Case Western Reserve University, Cleveland, OH 44106, USA. [10] Université de Montréal, Faculty of Medicine, Department of Microbiology, Infectiology and Immunology, Montréal H3C3J7 QC, Canada. Correspondence and requests for materials should be addressed to N.C. (email: nicolas.chomont@umontreal.ca)

Latently infected cells carrying integrated human immuno-deficiency virus (HIV) genomes persist during antiretroviral therapy (ART) and represent the main barrier to a cure[1–3]. The establishment of latency may result from direct infection of resting CD4+ T cells[4] or from infection of CD4+ T cells transitioning from an activated to a resting state[5]. Latently infected CD4+ T cells are rare both before and after ART initiation[6,7], suggesting that HIV latency is established only in a small fraction of CD4+ T cells.

Programmed cell death-1 (PD-1) is an immune checkpoint molecule expressed at high levels on the surface of exhausted HIV-specific CD4+ and CD8+ T cells[8–12]. Its blockade enhances CD4+ T cells and CD8+ T cells functions during Simian immunodeficiency virus infection[13,14].

In addition to its role in T-cell exhaustion, PD-1 and other immune checkpoint molecules such as lymphocyte activation gene-3 (LAG-3) and T-cell immunoreceptor with Ig and ITIM domains (TIGIT) are preferentially expressed at surface of persistently infected CD4+ T cells[15–17]. Of note, follicular helper T (Tfh) cells, which express high levels of PD-1, are major producers of viral particles in untreated HIV infection[18] and serve as a preferential reservoir for HIV during ART[19,20]. In addition, PD-1 and LAG-3 measured prior to ART strongly predict time to return of viraemia upon treatment interruption[21]. However, whether these molecules play an active role in the establishment and maintenance of HIV latency remains unclear.

In an in vitro latency model, PD-1 blockade reduces the frequency of latently infected CD4+ T cells[22]. Because PD-1 induces T-cell quiescence and inhibits T-cell activation[23], we hypothesized that the engagement of the PD-1 pathway may directly contribute to the establishment of viral latency by inhibiting viral transcription and production. We demonstrate that the engagement of PD-1 abrogates T-cell receptor (TCR)-induced HIV reactivation in latently infected cells isolated from HIV-infected individuals. Conversely, PD-1 blockade with the monoclonal antibody pembrolizumab enhances HIV production induced by the latency reversing agent bryostatin without increasing T-cell activation. These results suggest that the administration of immune checkpoint blockers to HIV-infected individuals on ART may facilitate latency reversal in vivo.

## Results

**PD-1 marks HIV-infected cells in viremic individuals**. To determine if PD-1 could play a role in the establishment of HIV latency, we first assessed the distribution of HIV in memory CD4+ T cells expressing high and low levels of PD-1 in HIV-infected individuals not receiving ART. We found that memory CD4+ T cells expressing PD-1 were preferentially infected, as demonstrated by the higher frequency of integrated HIV DNA in PD-1 expressing central (TCM), transitional (TTM), and effector memory (TEM) cells as compared to their PD-1 negative (PD-1−) counterparts (median fold-change: 6.5, 2.3, and 2.2, respectively, Supplementary Fig. 1a). Accordingly, flow cytometry sorted PD-1 positive (PD-1+) cells produced higher levels of viral particles, indicating that PD-1+ cells are major targets for productive HIV infection during untreated disease (Supplementary Fig. 1b).

**PD-1 engagement inhibits viral production**. To determine the impact of PD-1 engagement on HIV production, we stimulated productively infected CD4+ T cells isolated from untreated HIV-infected individuals in the presence or absence of PD-L1, one of the two ligands for PD-1. TCR stimulation led to a marked increase in the amount of the viral protein p24 measured in the culture supernatant and this induction was dramatically reduced in the presence of PD-L1 (98% inhibition, $p < 0.0001$, Fig. 1a and

Supplementary Fig. 1c). Of note, this effect was maintained over time, since the inhibition was still observed after 6 and 9 days of culture (Fig. 1b, c and Supplementary Fig. 1d). Importantly, PD-L1 acted directly on productively infected cells and not by decreasing the permissiveness of bystander cells since the inhibition of viral production was also observed in the presence of antiretroviral drugs (ARVs) that block new infection events as well as in short-term cultures (69–86% inhibition, Supplementary Fig. 1e–g). To demonstrate the specificity of this mechanism, we sorted PD-1+ and PD-1− memory CD4+ T cells and observed that PD-L1− mediated inhibition was restricted to PD-1+ cells ($p = 0.01$), since the ligand did not significantly affect viral production in PD-1− cells (Fig. 1d).

To gain further insights into the mechanism by which PD-1 engagement inhibited HIV production, we transfected primary CD4+ T cells with an LTR-luciferase reporter construct. The engagement of PD-1 by PD-L1 resulted in a significant reduction in the LTR activity, indicating that PD-1 exerted its inhibitory effect on HIV production at the transcriptional level ($p = 0.002$, Fig. 1e).

**PD-1 inhibits viral reactivation in latently infected cells**. Having demonstrated that the engagement of PD-1 at the surface of productively infected cells inhibited the viral production induced by TCR stimulation, we next sought to determine if this phenomenon could abrogate viral reactivation from latently infected cells. Even after years of ART, PD-1 expression at the surface of CD4+ T cells from virally suppressed individuals was not normalized (Supplementary Fig. 2a), notably in the TTM and TEM subsets (Supplementary Fig. 2b–e). Similarly, both PD-1 ligands (PD-L1 and PD-L2) were expressed at higher levels at the surface of CD4+ T cells and monocytes from fully suppressed individuals when compared to uninfected controls (Supplementary Fig. 2f–i). Of note, higher frequencies of CD4+ T cells expressing PD-1 were detected in the rectum, colon, and terminal ileum of individuals on ART when compared to matched blood and lymph nodes (Supplementary Fig. 2j). These higher frequencies of CD4+ T cells expressing PD-1 was consistent with heightened frequencies of differentiated memory CD4+ T cells (TTM and TEM) in the gut compartment (Supplementary Fig. 2k). We further confirmed our and others previous observations[16,19,24] indicating a strong positive correlation between the frequency of persistently infected CD4+ T cells (as measured by integrated HIV DNA) and PD-1 expression in CD4+ T cells during ART ($r = 0.42$, $p < 0.0001$, Supplementary Fig. 3a) as well as the preferential infection of PD-1+ TCM and TTM cells during ART by measuring integrated HIV DNA (median fold-enrichment: 1.7, $p = 0.01$ and 1.6, $p = 0.04$, respectively, Supplementary Fig. 3b). In addition, using the tat/rev induced limiting dilution assay (TILDA) and quantitative viral outgrowth assay, we observed that inducible and replication-competent proviruses were enriched in CD4+ T cells expressing PD-1 in the majority of the samples tested (Supplementary Fig. 3c, d).

To assess the effect of PD-1 engagement on viral reactivation from latency, we stimulated CD4+ T cells from virally suppressed individuals in the presence or absence of PD-L1. TCR stimulation led to readily detectable levels of viral production that were dramatically reduced in the presence of PD-L1 (89% inhibition, $p < 0.0001$, Fig. 2a), demonstrating that PD-1 engagement inhibits TCR-induced reactivation in latently infected CD4+ T cells. This inhibition was observed at the transcriptional level and restricted to PD-1+ cells (Supplementary Fig. 4a, b). Similarly, PD-1 engagement inhibited TCR-induced activation of the positive transcription elongation factor b (P-TEFb), a master regulator of HIV transcription (82% inhibition of CDK9 phosphorylation,

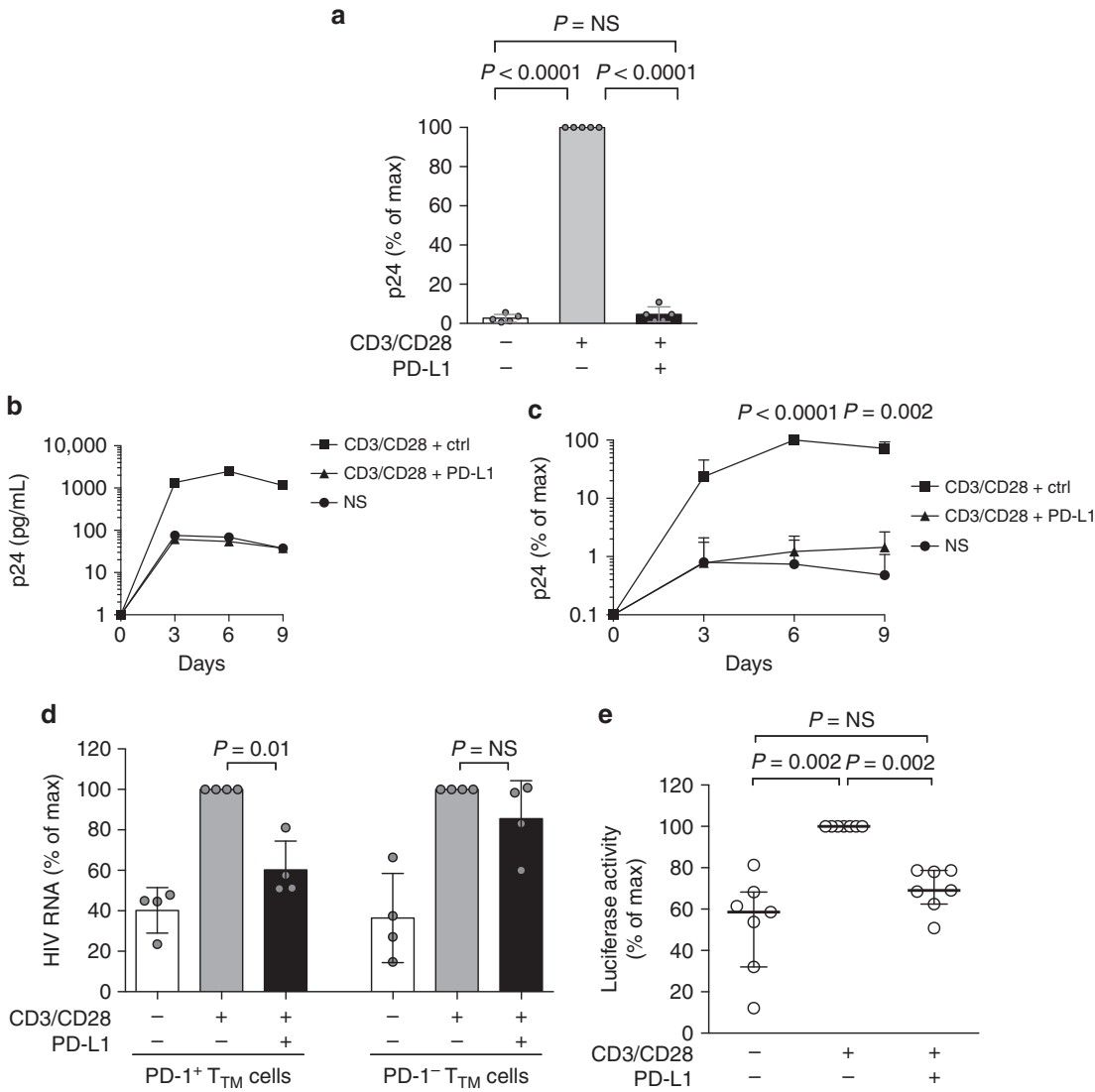

**Fig. 1** PD-1 engagement inhibits viral production in CD4[+] T cells from untreated HIV-infected individuals. **a** Relative viral production measured by p24 release in 3-days culture supernatants of CD4[+] T cells isolated from HIV-infected untreated individuals and stimulated through the TCR in the presence or absence of PD-L1 (means and standard deviations from $n = 5$ donors). $p$ Values were obtained from paired $t$ test analysis. **b** Same as in **a** with p24 measurements at day 3, 6, and 9 in CD4[+] T cells supernatants from a representative donor. **c** Relative viral production measured by p24 as in **b** (means and standard deviations from $n = 5$ donors). $p$ Values reflect differences between the PD-L1 and isotype control conditions and were obtained from paired $t$ test analysis. **d** Viral production normalized to the CD3/CD28 condition measured by RT-PCR in supernatants of sorted PD-1[+] and PD-1[−] $T_{TM}$ cells subjected to stimulation as in **a** (means and standard deviations from $n = 4$ donors). $p$ Values were obtained from paired $t$ test analysis. **e** Luciferase activity (normalized to the CD3/CD28-isotype ctrl condition) in CD4[+] T cells transfected with an LTR-luciferase reporter construct and stimulated as in **a** (means and standard deviations from $n = 7$ independent experiments). $p$ Values were obtained from paired $t$ test analysis. Source data are provided as a Source Data file

$p = 0.01$, Fig. 2b and 61% inhibition of cyclin T1 expression, $p = 0.02$, Fig. 2c)[25].

**PD-1 blockade enhances viral production ex vivo.** Since our results indicated that PD-1 engagement prevents HIV reactivation from latency, we sought to determine if PD-1 blockade could facilitate latency reversal. We stimulated large numbers of CD8- and CD56-depleted peripheral blood mononuclear cells (PBMCs) from HIV-infected individuals on suppressive ART with the latency reversing agent bryostatin in the presence of a PD-1 blocking antibody (pembrolizumab) or the corresponding isotype control. HIV production was greatly increased by PD-1 blockade (median fold-increase over isotype control

(interquartile range (IQR)) 36.00 (7.90–47.25), Fig. 2d) without enhancing T-cell activation nor affecting cell viability (Fig. 2e). Of note, blocking PD-1 also enhanced viral production when latently infected CD4[+] T cells were stimulated with suboptimal concentrations of the superantigens Staphylococcal Enterotoxins A and B (SEA and SEB) (median fold-increase over isotype control (IQR) 4.23 (2.04–71.96), Supplementary Fig. 5a). Interestingly, PD-1 blockade significantly increased the levels of cytokine production (Supplementary Fig. 5b, c) but had minimal effects on markers of T-cell homeostasis, proliferation, survival and activation upon TCR or IL-7 stimulations (Supplementary Fig. 6a–g). Together, these experiments indicated that blocking PD-1 increases viral production from latently

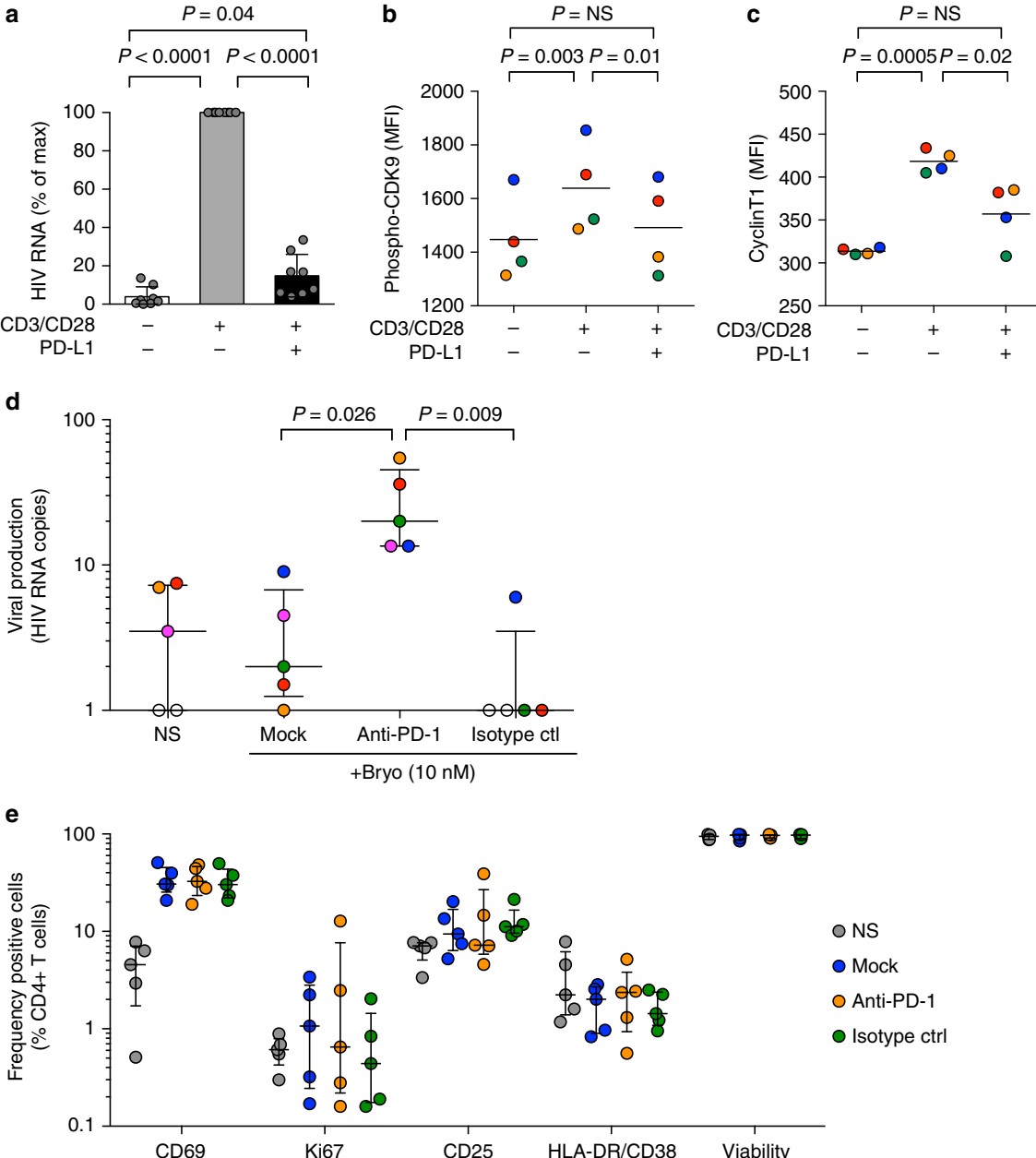

**Fig. 2** PD-1 engagement inhibits viral reactivation from latently infected cells and its blockade enhances viral production ex vivo. **a** Relative viral production measured by RT-PCR in 3-days culture supernatants of CD4[+] T cells isolated from virally suppressed individuals and stimulated through the TCR in the presence or absence of PD-L1 (means and standard deviations from $n = 8$ donors). $p$ Values were obtained from paired $t$ test analysis. **b** Level of CDK9 phosphorylation (Ser175) expression in CD4[+] T cells isolated from virally suppressed individuals after 12 h of stimulation through the TCR in the presence or absence of PD-L1 (black bars represent average MFI from $n = 4$ donors). $p$ Values were obtained from paired $t$ test analysis. **c** Level of cyclinT1 expression as in **b** (black bars represent average MFI from $n = 4$ donors). $p$ Values were obtained from paired $t$ test analysis. **d** Viral production measured by RT-PCR in 6-days culture supernatants of CD8- and CD56-depleted PBMCs isolated from virally suppressed individuals and stimulated with bryostatin (18 h pulse at 10 nM) in the presence of PD-1 blocking monoclonal antibody or the appropriate isotype control (10 μg/mL) (median and interquartile range from $n = 5$ donors). $p$ Values were obtained from paired t-test analysis of the log transformed value +1. **e** CD4[+] T cells activation measured by CD69, Ki67, CD25, and HLA-DR/CD38 expressions and cell viability measured by flow cytometry after 6 days of culture as described in Fig. 2d. Source data are provided as a Source Data file

infected cells in response to the latency reversing agent bryostatin and to suboptimal TCR stimulation without enhancing T-cell activation.

Our in vitro findings were further supported by the progressive decrease in total and integrated HIV DNA as well as in cell-associated HIV RNA (4-, 3-, and 19-fold decreases, respectively)

observed in a virally suppressed individual who received pembrolizumab for metastatic melanoma (Supplementary Fig. 7). We acknowledge that this case report does not prove latency reversal in vivo since a reduction in HIV persistence markers could result from multiple mechanisms, including immune-mediated effects. Nonetheless, this observation supports the

concept that PD-1 blockade can reduce the latent reservoir in vivo.

## Discussion

Latently infected cells represent the major obstacle to a cure for HIV infection[26]. Memory CD4[+] T cells are constantly exposed to homeostatic signals required for their maintenance (IL-7, IL-15, and perhaps interactions with self-peptide MHCII complex). However, the long half-life of memory CD4[+] T cells and the modest activity of the majority of latency reversing agents[27] suggest that inhibitory mechanisms may continuously abrogate viral reactivation in latently infected cells. Our results indicate that HIV preferentially persists in PD-1[+] cells and that the engagement of PD-1 potently inhibits viral reactivation. It was recently reported that PD-1 suppresses T-cell function primarily by inactivating CD28 signaling[28,29], which provides a potential explanation for the preferential persistence of HIV in PD-1[+] $T_{CM}$ and $T_{TM}$ cells, two subsets that express high levels of CD28.

The ability of TCR signaling to enhance HIV transcription has been associated to release of P-TEFb from an inactive complex through an ERK-dependent pathway[30]. In this study, we provide mechanistic evidence that PD-1 engagement inhibits the P-TEFb signaling pathway, which likely contributes to the inhibition of HIV production. In addition to its immediate effect on HIV production, PD-1 engagement may also lead to a prolonged silencing of the HIV promoter through metabolic and epigenetic reprogramming of T cell[31–33].

Importantly, our observations demonstrate that HIV reactivation is enhanced by PD-1 blockade in CD4[+] T cells from virally suppressed individuals ex vivo. Of note, we observed that PD-1 blockade potentiates latency reversal without inducing global T-cell activation, in spite of increased levels of TCR-induced cytokine production. These observations suggest that cytokine secretion and HIV reactivation can be uncoupled from the induction of T-cell activation. This is in line with studies reporting that reduced levels of IL-2 production is sensitive hallmark of PD-1 engagement. Since NF-κB is an important factor positively regulating both IL-2 and HIV transcription, it is possible that activation of this transcription factor is involved in the increased levels of viral production following PD-1 blockade. As such, increased IL-2 secretion following PD-1 blockade was also reported in vivo[34]. Since the level of PD-1 expression on distinct CD4[+] T-cell subsets impacts the effect of PD-1/PD-L1 blockade[11], it remains to be determined if HIV latency can be reverted in all cellular HIV reservoirs using this strategy. Of note, since additional co-inhibitory receptors have been associated with HIV persistence in different subsets[16,17,35] combination of immune-checkpoint blockers may be necessary to achieve optimal latency reversal.

We observed a progressive decline of several markers of HIV persistence throughout treatment with pembrolizumab in an individual receiving this antibody for malignant melanoma. This case illustrates the ability of PD-1 blockade to reduce the size of the HIV reservoir in some individuals, which is consistent with other studies demonstrating an effect of PD-1 or CTLA-4 blockade on HIV reservoir dynamics[22,36,37]. Since this effect is not observed in all virally suppressed individuals[38], results from larger ongoing studies enrolling HIV-infected participants receiving immunotherapy for the treatment of malignancies are warranted.

The potential toxicity of these immunotherapies and the risk benefit ratio for HIV-infected individuals on stable suppressive ART should be considered carefully. To our knowledge, safety data in HIV-infected noncancer patients are not yet available. However, several reports in individuals with HIV and cancer[38–40]

indicate that the frequency of immune related adverse events due to PD-1 blockade in the setting of HIV appears to be similar to non-HIV. Of note, in all these studies including ours, the impact of PD-1 blockade on the HIV reservoir was assessed in peripheral blood. Knowing that CD4[+] T cells expressing PD-1 are found at high frequencies in preferential anatomical HIV reservoirs such as gut, lymph nodes and adipose tissue during ART[19,41,42], assessing latency reversal and HIV reservoir dynamics after PD-1 blockade administration would require analyzing tissues in which this process is more likely to occur and collecting samples shortly after administration of the PD-1 blocking antibody.

To conclude, our findings suggest that PD-1 may provide latently infected cells with a selective advantage to persist during ART by continuously inhibiting viral reactivation and provide the rationale for evaluating immune checkpoint blockers in curative strategies.

## Methods

**Study population**. HIV-seropositive individuals on stable suppressive ART, chronically infected individuals with no history of ART and uninfected controls enrolled in this study. All participants signed inform consent approved by the Martin Memorial Health Systems (FL, USA), the UCSF (CA, USA), the Royal Victoria Hospital and the CHUM hospital (QC, Canada) review boards (IRB #10-1320, Ref # 068192 and FWA #00004139, respectively). The participant with metastatic malignant melanoma who received pembrolizumab-based immunotherapy was virally suppressed for more than 3 years and signed inform consent. Blood was collected 8 days after the second administration of the antibody and 20 to 40 days after all subsequent treatment cycles over a period of 4 months. We complied with all relevant ethical regulations for work with human participants.

**Immunophenotyping**. PBMCs were isolated from peripheral blood and leukapheresis using Ficoll–Paque density gradient centrifugation. Single-cell suspensions from lymph node, rectum, colon, and ileal biopsies were prepared as previously described[41,43]. Cryopreserved PBMCs were thawed, washed and stained for phenotyping or cell sorting. To measure the expression of PD-1 in subsets of memory CD4[+] T cells, the following antibody panel was used: CD3-Alexa700 (clone UCHT1, BD#557943), CD4-QDot605 (clone S3.5, Invitrogen#Q10008), CD8-PB (clone RPA-T8, BD#558207), CD14-V500 (clone M5E2, BD#561391), CD19-AmCyan (clone SJ25C1, BD#339190), LIVE/DEAD Aqua marker (Invitrogen#L34957), CD45RA-APC-H7 (clone HI100, BD#560674), CD27-BV650 (clone O323, Biolegend#302828), and CCR7-PE-Cy7 (clone 3D12, BD#557648) and PD-1-APC (clone MIH4, eBioscience#17-9969-42). Gates were defined using fluorescence minus one controls. CD4[+] T-cell subsets were identified by CD27, CD45RA, and CCR7 expression on CD4[+] T cells after exclusion of dump positive cells (LIVE/DEAD, CD14 and CD19). PD-1 was measured in gated CD4[+] T-cell subsets including naïve CD4[+] T cells (CD3+ CD8-CD4+ CD45RA+ CCR7+ CD27+), central memory CD4[+] T cells (CD3+ CD8− CD4+ CD45RA− CCR7+ CD27+), transitional memory CD4[+] T cells (CD3+ CD8− CD4+ CD45RA− CCR7− CD27+), effector memory CD4[+] T cells (CD3+ CD8-CD4+ CD45RA− CCR7− CD27−) and terminally differentiated CD4[+] T cells (CD3+ CD8-CD4+ CD45RA+ CCR7− CD27−). To measure the expression of PD-1 ligands (PD-L1 and PD-L2) the following antibody panel was used: CD3-PB (clone UCHT1, BD#558117), CD4-Alexa700 (cloneRPA-T4, BD#557922), CD8-PerCPCy5.5 (clone RPA-T8, BD#560662), CD14-FITC (clone M5E2, BD#555397), LIVE/DEAD Aqua marker (Invitrogen#L34957), CD45RA-APC-H7 (clone HI100, BD#560674), PD-1-APC (clone MIH4, eBioscience#17-9969-42), PD-L1-PE-Cy7 (clone MIH1, BD#558017), and PD-L2-PE (clone MIH18, BD#558066). Data were acquired on a BD LSR II flow cytometer using the FACSDiva software (Becton Dickinson) and analyzed using Flow Jo version 9 (Treestar).

**Cell sorting**. Central, transitional and effector memory CD4[+] T cells were sorted based on their expression of PD-1. The antibodies used for sorting were similar than those used for phenotyping (Supplementary Fig. 8). In a second set of experiments, total memory CD4[+] T cells (CD3+ CD4+ CD45RA−) were sorted based on their expression of PD-1. Sorted cells were collected using a FACSAria II cell sorter (Becton Dickinson).

**Isolation of total CD4[+] T cells**. Total CD4[+] T cells were isolated from cryopreserved PBMCs using magnetic negative selection as per the manufacturer's protocol (Stem Cell Technologies, Vancouver, Canada).

**Quantification of integrated HIV DNA**. Total CD4[+] T cells or sorted CD4[+] T-cell subsets were used to measure the frequency of cells harboring integrated HIV DNA by real time nested polymerase chain reaction (PCR)[44]. Briefly, cells were lysed by proteinase K digestion. Cell lysates were directly used for HIV DNA

quantifications. Integrated HIV-1 DNA was amplified with two Alu primers together with a primer annealing the LTR/gag region (Supplementary Table 1). In all PCR reactions, primers specific for the CD3 gene were added to precisely quantify the cell input. In a second round of PCR, appropriate primers and probes were used to amplify HIV sequences from the first round of amplification. Inner primers specific for the CD3 gene were used in a separate reaction to determine cell input. The number of copies of integrated HIV-1 DNA was calculated by using serial dilutions lysed ACH-2 cells as a standard curve. Results were expressed as numbers of HIV copies per million cells.

**Quantification of Tat/rev inducible multiply spliced HIV RNA**. The frequency of CD4[+] T cells with inducible multiply spliced HIV RNA was determined using the TILDA[6]. Briefly, cells were stimulated with phorbol myristate acetate (PMA; 100 ng/mL) and ionomycin (1 μg/mL) for 12 h. Serial dilutions of the stimulated cells (18,000; 9000; 3000, and 1000 cells; 24 replicates per dilution) were distributed in a 96 well plate containing real time PCR (RT-PCR) buffer. Multiply spliced HIV RNA was quantified by semi nested real time PCR using primers specific for the tat/rev region (Supplementary Table 1). The frequency of positive cells was calculated using the maximum likelihood method and this number was then expressed as a frequency of cells with inducible msHIV RNA per million cells.

**Quantification of cell-associated and cell-free HIV RNA**. Cell-associated RNA were extracted using the RNeasy kit (Qiagen). Alternatively, freshly collected cell culture supernatants were ultracentrifuged for 1 h at 25,000g to pellet HIV particles. By using this procedure, only packed viral RNAs were quantified, excluding HIV RNAs that are passively released in the medium as a consequence of cell death. Viral RNAs were extracted using the Qiamp viral RNA kit (Qiagen) and quantified using an ultrasensitive semi-nested real time RT-PCR with a detection limit of a single copy of HIV RNA per reaction[45].

**Quantification of viral production by p24 ELISA**. Viral production was measured by quantification of the p24 antigen by a sensitive in-house p24 ELISA[46]. In this enzymatic assay, 183-H12-5C (coating) and 31-90-25 (conjugated to biotin) antibodies were used in combination to quantify p24 levels.

**Quantification of replication-competent virus**. Sorted memory PD1[+] and PD1[−] CD4[+] T cells were activated for 3 days with coated anti-CD3 (clone OKT3) and soluble anti-CD28 (clone CD28.2) in presence of anti-PD-1 antibody (pembrolizumab 10 μg/mL) in limiting dilution (from 1.5 million cells to 0.05 million cells). After 3 days of stimulation, cells were washed 2 times with RPMI and co-cultured for an additional 12 days with PHA/IL-2 activated CD4[+] T cells from two HIV negative donors at a ratio 1:3 in medium containing IL-2 (5 ng/mL) and anti-PD-1 antibody (pembrolizumab 10 μg/mL, Merck). Culture were split twice weekly; half of cell culture supernatants were collected for quantification of the viral production by p24 ELISA. Infectious Unit per Million cells was determined based on the number of wells positive for p24 using the maximum likelihood method (http://silicianolab.johnhopkins.edu/).

**PD-1 engagement**. CD4[+] T cells were cultured in RPMI 1640 complemented with 10% fetal bovine serum and 1% penicillin–streptomycin in the presence or absence of ARVs [100 nM efavirenz (EFV), 180 nM zidovudine (AZT), and 200 nM raltegravir (RAL)]. Cells were stimulated with CELLection Pan Mouse IgG (Invitrogen) coated with anti-CD3 (UCHT1), anti-CD28 (CD28.2), and PD-L1 chimeric protein (kindly provided by G.J. Freeman) or the appropriate isotype control (IgG2a). Beads were added to CD4[+] T cells in a 1:2 (cell:bead) ratio and kept in culture for the entire duration of the experiment (up to 9 days).

**PD-1 blockade**. Fifteen to 25 × 10^6 CD8-depleted PBMCs or CD56- and CD8-depleted PBMCs from virally suppressed individuals were stimulated with SEA/SEB (0.3 ng/mL), IL-7 (10 ng/mL), or bryostatin (10 nM, 18 h exposure) in the presence or absence of anti-PD-1 (pembrolizumab 10 μg/mL) or the appropriate isotype control, in the presence of ARVs. Viral production was measured in cell culture supernatants after 3 or 6 days of culture as described above. Samples with spontaneous viral production (>10 HIV RNA copies in the nonstimulated condition) and/or inconsistent replicate values attributed to low frequency of reservoir cells were excluded. Cytokine production was measured by electro-chemiluminescence using multi-array technology (Meso Scale Discovery). Bcl-2, CD69, Ki67, CD25, and HLA-DR/CD38 expressions, phosphorylation of STAT5 and Akt and cell viability were measured by flow cytometry using the following antibodies: Bcl-2-V450 (clone Bcl-2/100, BD#560637), CD69-APC (clone FN50, BD#555533), CD25-BUV737 (clone 2A3, BD#564385), HLA-DR-BUV395 (clone G46-6, BD#564040), CD38-PE (clone HIT2, BD#555460), pSTAT5-AF647 (pY694, BD#612599), and pAkt-PE-CF594 (pS473, BD#562465). PD-1 receptor occupancy by the blocking antibody was measured in all experiments by flow cytometry and was constantly higher than 90%.

**P-TEFb activity**. Phosphorylation of CDK9 (Ser175) (provided by Merck) and level of expression of CyclinT1 were measured by flow cytometry after 12 h

stimulation of CD4[+] T cells isolated from virally suppressed individuals upon PD-1 engagement[25]. Briefly, titrated antibodies (CD3-FITC, CD4-AF700, CD8-PB, CD45RA-APC-H7, PD-1-BUV737) were added to 2 million cells in 50 μl of PBS in the presence of Live/Dead Dye for 30 min at 4 °C. Cells were fixed with Formaldehyde 4% (methanol free) for 20 min at room temperature and then permeabilized with Perm/Wash Buffer (BD#554723) for 30 min at 4 °C. Intracellular stainings for pCDK9 (pCDK9-AF647) and CyclinT1 (CyclinT1-TRITC, Santa-Cruz#8127) were conducted in Perm/Wash Buffer for 20 min at room temperature.

**LTR-luciferase**. Memory CD4[+] T cells were isolated by negative magnetic selection (StemCell) from fresh PBMCs obtained from HIV-negative individuals. After 24 h of TCR stimulation (0.1 μg/mL coated anti-CD3 and 1 μg/mL soluble anti-CD28), cells were nucleofected with pBlue3′LTR-lucB (NIH AIDS Reagent Program) using Amaxa technology (Lonza). Twenty-four hour postnucleofection, cells were stimulated with CD3/CD28 ± PD-L1 beads, as described above. Luciferase activity was measured after 6 h of stimulation.

**Statistical analysis**. We performed Pearson's correlation, t test and Mann–Whitney U tests with Prism 6.0 software. We considered two-sided p values of less than 0.05 significant.

**Reporting summary**. Further information on experimental design is available in the Nature Research Reporting Summary linked to this article.

## Data availability

The data that support the findings of this study are available from the corresponding author upon reasonable request. Source data underlying all figures are provided as a Source Data file.

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

## Acknowledgments

The authors thank the participants who have generously donated samples for this study. The authors thank Moti Ramgopal, Brenda Jacobs, Mario Legault, Guillaume Thériault, Catalin Mihalcioiu, and Josée Girouard for recruitment and clinical assistance with study participants, Gordon Freeman for providing the PD-L1 chimeric protein, Sheri Dubey and Danilo Casimiro for cytokines quantifications, Jonathan Karn, Curtis Dobrowolski, Marion Pardons for assistance with P-TEFb measurements and Mariam Lawani, Wendy Picano, and Hawley Rigsby for technical assistance. We also thank the flow cores at UCSF, VGTI Florida (Yu Shi and Kim Kusser) and at the CRCHUM (Dominique Gauchat and Annie Gosselin) for cell sorting and the NC3 core at the CRCHUM (Olfa Debbeche). The authors thank Marta Massanella and the members of the Cleveland Immunopathogenesis Consortium for advice and helpful discussions. This work was supported the Delaney AIDS Research Enterprise (DARE) to Find a Cure 1U19AI096109 and UM1AI126611, NIH grant 1R21AI113096, by the Foundation for AIDS Research (amfAR Research Consortium on HIV Eradication 108687-54-RGRL and 108928-56-RGRL), by the Canadian Institutes for Health Research (#364408), by The Canadian HIV Cure Enterprise Team Grant (Cancure) from the CIHR in partnership with CANFAR and IAS (HIG-133050), and by the Réseau SIDA maladies infectieuses (FRQ-S). The SCOPE cohort was supported by the UCSF/Gladstone Institute of Virology & Immunology CFAR (P30 AI027763) and the CFAR Network of Integrated Systems (R24 AI067039). Additional support was provided by the Delaney AIDS Research Enterprise (DARE; AI096109, A127966) and the amfAR Institute for HIV Cure Research (amfAR 109301). S.R.L. is supported by a practitioner fellowship from the National Health and Medical Research Council (NHMRC) of Australia. J.P.R. holds the Louis Lowenstein Chair in Hematology & Oncology in the Faculty of Medicine at McGill University. N.C. is supported by a Research Scholar Career Award of the Quebec Health Research Fund (FRQ-S, #30950).

## Author contributions

R.F., R.P.S. and N.C. designed the studies. R.F., S.D.F., M.E.F., F.A.P., and N.C. performed the experiments. R.F., S.D.F., R.P.S. and N.C. analyzed the data. C.T.C., F.M.H., R.H., S.G.D. and J.P.R. recruited participants. S.R.L., S.G.D. and D.J.H. provided critical intellectual contributions. R.F., S.D.F. and N.C. wrote the manuscript with contributions from all co-authors.

## Additional information

**Competing Interests:** D.J.H. is an employee and shareholder of Merck & Co., Inc., manufacturer of pembrolizumab. S.G.D. has received grant support from Gilead, Merck, and ViiV. He has consulted from AbbVie, Janssen and Shionogi. He is a member of the scientific advisory boards for Enochian Biosciences and BryoLogyx. N.C. has received grant support from Merck. The remaining authors declare no competing interests.

