## [Peer Review File · Nature Communications]

Reviewers' Comments:

Reviewer #2:

Remarks to the Author:

This is a very timely study addressing a key question for the HIV cure field, specifically, if expression and signaling through PD-1 contributes to the establishment and maintenance of viral persistence during ART treatment. Several high impact publications showed that HIV and SIV preferentially persists in memory CD4 T cells expressing PD-1 and/or other co-inhibitory receptors (LAG-3, CTLA-4, TIGIT), as well as that the frequency of CD4 T cells expressing these receptors prior to ART predict time to viral rebound upon ART treatment interruption. However, it is still unclear if this is only an association or PD-1 (and others co-inhibitory receptors) plays a direct role in promoting the establishment and maintenance of the latent HIV reservoir. In my opinion, the manuscript convincingly proves, with a set of well designed, controlled, and interpreted ex vivo experiments, an active role for PD-1 signaling. The authors first demonstrated that PD-1 engagement inhibits viral production and prevents HIV reactivation from latently infected cells; then, they proved that PD-1 blockade (in vitro) can potentiate LRA in reversing HIV latency in CD4 T cells from ART-treated HIV infected individuals. The manuscript is well written, the conclusions strongly supported by the findings and balanced.

I previously reviewed this manuscript for NMED. All my previous comments (including mentioning the role of others co-inhibitory receptors, the likely need of multiple blockades, the potential toxicity of these interventions) have been addressed. As requested, I went in details through the previous Reviewer 1 comments and I do believe the authors addressed all of them. The data in one patient receiving anti-PD-1 in vivo is interesting and supportive of the overall proposed model. The authors highlighted this is limited to one individual, quoted other studies with similar results, and correctly stated that larger studies are needed to determine how PD-1 blockade can impact on HIV persistence in vivo. Considering I favorably reviewed this manuscript already for NMED, the authors addressed all my comments, and answered reviewer 1 main concerns, I have only a minor comment to add, detailed below.

Minor comment:

A study in Rhesus Macaques infected with SIV recently reported that administration of a PD-1 blocking antibody during ART transiently increases plasma viremia, suggesting an anti-latency effect of PD-1 blockade in vivo (Mylvaganam, JCI Insight 2018).

In the quoted study, a similar increased in plasma viremia was observed also in ART-only controls. The levels of viral reactivation were not significantly higher in animals receiving anti-PD-1 as compared to controls. As such, that sentence should be removed.

REVIEWERS' COMMENTS:

Reviewer #2 (Remarks to the Author):

This is a very timely study addressing a key question for the HIV cure field, specifically, if expression and signaling through PD-1 contributes to the establishment and maintenance of viral persistence during ART treatment. Several high impact publications showed that HIV and SIV preferentially persists in memory CD4 T cells expressing PD-1 and/or other co-inhibitory receptors (LAG-3, CTLA-4, TIGIT), as well as that the frequency of CD4 T cells expressing these receptors prior to ART predict time to viral rebound upon ART treatment interruption. However, it is still unclear if this is only an association or PD-1 (and others co-inhibitory receptors) plays a direct role in promoting the establishment and maintenance of the latent HIV reservoir. In my opinion, the manuscript convincingly proves, with a set of well designed, controlled, and interpreted ex vivo experiments, an active role for PD-1 signaling. The authors first demonstrated that PD-1 engagement inhibits viral production and prevents HIV reactivation from latently infected cells; then, they proved that PD-1 blockade (in vitro) can potentiate LRA in reversing HIV latency in CD4 T cells from ART-treated HIV infected individuals. The manuscript is well written, the conclusions strongly supported by the findings and balanced.

I previously reviewed this manuscript for NMED. All my previous comments (including mentioning the role of others co-inhibitory receptors, the likely need of multiple blockades, the potential toxicity of these interventions) have been addressed. As requested, I went in details through the previous Reviewer 1 comments and I do believe the authors addressed all of them. The data in one patient receiving anti-PD-1 in vivo is interesting and supportive of the overall proposed model. The authors highlighted this is limited to one individual, quoted other studies with similar results, and correctly stated that larger studies are needed to determine how PD-1 blockade can impact on HIV persistence in vivo. Considering I favorably reviewed this manuscript already for NMED, the authors addressed all my comments, and answered reviewer 1 main concerns, I have only a minor comment to add, detailed below.

We thank the reviewer for his positive evaluation of our manuscript.

Minor comment:

A study in Rhesus Macaques infected with SIV recently reported that administration of a PD-1 blocking antibody during ART transiently increases plasma viremia, suggesting an anti-latency effect of PD-1 blockade in vivo (Mylvaganam, JCI Insight 2018). In the quoted study, a similar increased in plasma viremia was observed also in ART-only controls. The levels of viral reactivation were not significantly higher in animals receiving anti-PD-1 as compared to controls. As such, that sentence should be removed. The reviewer is correct. The sentence has been removed from the revised manuscript.